# Generalization from Starvation: Hints of Universality in LLM Knowledge Graph Learning

## Abstract

Motivated by interpretability and reliability, we investigate how neural networks represent knowledge during graph learning, We find hints of universality, where equivalent representations are learned across a range of model sizes (from $10^2$ to $10^9$ parameters) and contexts (MLP toy models, LLM in-context learning and LLM training). We show that these attractor representations optimize generalization to unseen examples by exploiting properties of knowledge graph relations (e.g. symmetry and meta-transitivity). We find experimental support for such universality by showing that LLMs and simpler neural networks can be stitched, i.e., by stitching the first part of one model to the last part of another, mediated only by an affine or almost affine transformation. We hypothesize that this dynamic toward simplicity and generalization is driven by "intelligence from starvation": where overfitting is minimized by pressure to minimize the use of resources that are either scarce or competed for against other tasks.

## 1 Introduction

Large Language Models (LLMs), despite being primarily trained for next-token predictions, have shown impressive reasoning capabilities (Bubeck et al., 2023; Anthropic, 2024; Team et al., 2023). However, despite recent progress reviewed below, it is not well understood what knowledge LLMs represent internally and how they represent it. Improving such understanding could enable valuable progress relevant to transparency, interpretability, fairness and robustness, for example

- discovering and correcting inaccuracies to improve model reliability.

- discovering and correcting bias,

- revealing and removing dangerous knowledge (relevant to bioweapon design, say),

- detecting deceptive behavior where models deliberately output information inconsistent with its knowledge.

The goal of this paper is to deepen our our understanding of learned knowledge representations by focusing specifically on the representations learned of *knowledge graphs* (KGs), which are loosely speaking a discrete set of entities with various relations between them. KGs provide a valuable test-bed because, although they are simpler and more structured than the totality of implicit knowledge LLM training corpora, they can nonetheless capture a massive amount of valuable human knowledge. We focus on applying the approach of mechanistic interpretability not to learned *algorithms*, but to learned *knowledge*.

The rest of this paper is organized as follows: We relate our approach to prior work in Section 2. In Section 3, we formally describe our problem settings, and Section 4 explains various methods of measuring representation alignment, including model stitching. Section 5 presents hints of universality via LLM stitching, and Section 6 takes a further glimpse into the geometrical structure of LLMs' representations. We discuss our starvation hypothesis for hints of universality in Section 7, and summarize our conclusions in Section 8.

## 2 RELATED WORK

Following the drastic enhancement of LLMs' capabilities, understanding the inner workings of Large Language Models have become increasingly important to ensure the safety and robustness of AI systems (Tegmark & Omohundro, 2023; Dalrymple et al., 2024).

**Mechanistic Interpretability** Neural Networks have demonstrated a surprising ability to generalize (Liu et al., 2021; Ye et al., 2021). Recently, there have been increasingly more efforts on trying to reverse engineer and interpret neural networks' internal operations (Zhang et al., 2021; Bereska & Gavves, 2024; Baek et al., 2024). Such methods include using structural probes and interventions at the level of entire representations (Hewitt & Manning, 2019; Pimentel et al., 2020), and studying neuron activations at the individual neuron level (Dalvi et al., 2019; Mu & Andreas, 2020). Our work contributes to these efforts, aiming to understand the knowledge representations in LLMs.

**Knowledge Representations in Language Models** Several studies have presented evidence for optimism about LLMs' intelligence, showing that LLMs are capable of forming conceptual representations in spatial, temporal, and color domains (Gurnee & Tegmark, 2023; Abdou et al., 2021; Li et al., 2021). Some studies focused primarily on examining the linearity of LLMs' feature representations (Gurnee & Tegmark, 2023; Hernandez et al., 2023). Several works have found multi-dimensional representations of inputs such as lattices (Michaud et al., 2024) and circles (Liu et al., 2022; Engels et al., 2024), one-dimensional representations of high-level concepts and quantities in large language models (Gurnee & Tegmark, 2023; Marks & Tegmark, 2023; Heinzerling & Inui, 2024; Park et al., 2024).

Language model representations could be viewed as extensions of early word embedding methods such as GloVe and Word2vec, which were found to contain directions in their vector spaces corresponding to semantic concepts, e.g. the well-known formula f(king) - f(man) + f(woman) = f(queen) (Drozd et al., 2016; Pennington et al., 2014; Ma & Zhang, 2015). Language model representations could also be viewed as a generalization of traditional knowledge graph embedding models such as TransE (Wang et al., 2014), ComplexE (Trouillon et al., 2016), and TransR (Lin et al., 2015), typically embed both entities and relations in the latent space, and optimize the score function to perform link prediction task.

**Representation Alignment and Model Stitching** There are active discussions in the literature about strengths and weaknesses of different representation alignment measures (Huh et al., 2024; Bansal et al., 2021; Sucholutsky et al., 2023). Several works have considered stitching to obtain better-performing models, such as stiching vision and language models for image and video captioning task (Li et al., 2019; Iashin & Rahtu, 2020; Shi et al., 2023), and stitching BERT and GPT for improved performance in look ahead section identification task (Jiang & Li, 2024). Some works have considered stitching toy transformers to understand the impact of activation functions on model's performance (Brown et al., 2023). Our work considers stitching LLMs to examine the hints of universality across scales, and to better understand the knowledge processing process of LLMs.

## 3 PROBLEM FORMULATION

In this section, we define our notation and terminology.

**Knowledge Graph Learning:** Consider a general knowledge graph (KG) consisting of $m$ binary relations $R^{(1)}, R^{(2)}, \cdots R^{(m)}$ between $n$ objects (nodes) $x_1, ..., x_n$. A KG simply generalizes the concept of a *directed graph* (a single binary relation $R(x_i, x_j) \mapsto \{0, 1\}$ with directed edges linking nodes $R(x_i, x_j) = 1$) by allowing multiple types of edges $R^{(i)}$ which we may imagine having different colors. Our machine-learning task is link prediction: to predict the probability $p_{ijk}$ that $R^{(i)}(x_j, x_k) = 1$ by training on a random data subset. If desired, we can further generalize from binary relations to allow relations that take fewer or more arguments (unary or ternary relations, say). A trivial example is where $x_i = i$, $m = 1$ and $R^{(1)}(x_i, x_j) = 1$ iff $i > j$. A richer example we will study involves family trees where each node $x_i$ is a person and there are relations such as "sister", "mother", "descendant" and "spouse".

**KG representations:** Many of the most popular KG-learning-algorithms embed both the relations $R_i$ and the elements $x_j$ in a vector space $\mathbb{R}^d$ and train a link prediction function $p_{ijk}$ of some clever

predetermined parametrized form (Cao et al., 2024). Neural networks learning link prediction "in the wild" have more freedom: although they typically learn node embeddings $x_i$ (both in the tokenization stage and as subsequent layers embed higher-level concepts), they have the freedom to learn custom link prediction algorithms for each relation. We therefore generalize the aforementioned approach by embedding only objects ($x_i \mapsto \mathbf{E}_i \equiv \mathbf{E}(x_i) \in \mathbb{R}^d$) and not relations, instead training a link predictor network $\mathbf{p}(\mathbf{E}_j, \mathbf{E}_k)$ to input two embedding vectors $\mathbf{E}_i, \mathbf{E}_j$ and output an $m$-dimensional vector $\mathbf{p}$ with a separate link probability prediction $p_k$ for each of the $m$ relations. What we term a KG *representation* is therefore a cloud of $n$ points (corresponding to objects) in the embedding space $\mathbb{R}^d$.

**Optimal KG representations:** Many relations have special properties that enable generalization to cases not seen during training. For example, if a relation $R$ is symmetric and transitive, then seeing $R(x_1, x_2) = 1$ and $R(x_2, x_3)$ in the training set implies that $R(x_2, x_1) = R(x_3, x_2) = R(x_1, x_3) = R(x_3, x_1) = 1$. We define a relationship *property* as an identity involving only logical operations, quantifiers and relations. For example, $R$ having the transitivity property corresponds to the identity $\forall i, j, k : R(x_i, x_j) \wedge R(x_j, x_k) \implies R(x_i, x_k)$. We also consider properties linking different relations. For example, if $R^{(1)} = parent$ and $R^{(2)} = grandparent$, then we have what we term *meta-transitivity*: $\forall i, j, k : R^{(1)}(x_i, x_j) \wedge R^{(1)}(x_j, x_k) \implies R^{(2)}(x_i, x_k)$.
**Definition:** *A combination of a KG representation and a predictor function is optimal if automatically ensures that all properties of the relations hold.*

For example, consider a KG consisting of nodes $x_i \in \mathbb{R}$ and the greater-than-relation $R^{(1)}(x_i, x_j) = 1$ iff $x_i > x_j$. The simple 1-dimensional embedding $\mathbf{E}(x_i) = \mathbf{x}_i$ combined with the predictor function $p(E_i, E_j) = H(E_i - E_j)$ where $H$ is the heaviside step function ($H(x) = 1$ if $x > 0$, vanishing otherwise) is optimal, since it guarantees antisymmetry and transitivity. A dumb 1-dimensional embedding mapping each $x_i$ into a random number, combined with a decoder that simply memorizes whether $R$ holds for each embedding pair, is non-optimal, since it merely memorizes/overfits the training data and has no ability to generalize to pairs not seen during training.

**Representation symmetries:** Given a KG representation and a predictor function, we define its *symmetry group $G$* as all invertible mappings $g$ of the embedded point cloud $\mathbf{E}_i$ that leaves the link predictions vector $\mathbf{p}(\mathbf{E}_i, \mathbf{E}_j)$ invariant. We will see that in many cases, we can find representations that are provably optimal and explicitly compute their symmetry group. For the simple aforementioned example of the greater-than relation, the symmetry group is any remapping of the embedded points $\mathbf{E}_i$ that preserves their ordering.

**Representation equivalence:** We define two representations as *equivalent* if an element of the symmetry group of one maps it into the other. For example, for the greater-than example with $n = 3$, the two embeddings $\{E_i\} = \{-3.14, 1.41, 3.14\}$ and $\{1, 2, 3\}$ are equivalent.

## 4 MEASURING REPRESENTATION ALIGNMENT VIA STITCHING

There are active discussions in the literature about strengths and weaknesses of different representation alignment measures (Huh et al., 2024; Bansal et al., 2021; Sucholutsky et al., 2023). In this paper, we quantify representation equivalence as the accuracy that one model's decoder gets on a trained affine transformation or almost affine transformation (AAT) of the output from another model's encoder. This AAT also enables visual comparison of the two learned representations in the same space. We refer to this Equivalence Score (ES) as AAT-ES. This is a generalization of model stitching, where a small quadratic term is introduced to absorb any superfluous nonlinearities that may emerge during nonlinear model training.

**Definition 1.** (AAT, Almost Affine Transformation) $f : \mathbf{E} \to \mathbf{G}$ is an AAT if there exists $\mathbf{b}, \mathbf{c}, \mathbf{d}$ such that $f(\mathbf{E}_i) = \mathbf{b}_i + \sum_k \mathbf{c}_k \mathbf{E}_{ik} + \epsilon \sum_{k,l} \mathbf{d}_{k,l} \mathbf{E}_{i,k} \mathbf{E}_{i,l}$, where a small quadratic term $\epsilon \ll 1$ is introduced to absorb any superfluous nonlinearities that may emerge during nonlinear model training.

While AAT-ES is a parametric measure that requires training (and hence more computational resources) as opposed to nearest-neighbor-based metrics such as CKA (Centered Kernel Alignment), we believe that AAT-ES is a better measure of representation alignment, as it directly compares the embedding with the decoder. Due to representation symmetries, there are many *degenerate* representations that can be perfectly decoded by the same decoder, but do not necessarily have the same

**Pythia410m + Pythia1.4b, Stitched at 20%**

```
Functional magnetic resonance imaging (fMRI) is
a noninvasive imaging technique that uses
magnetic fields to generate images of the heart.
```

**Opt2.7b + Opt6.7b, Stitched at 50%**

```
The shutdown of the island was planned and
orderly, but the chaos that followed the
shutdown was not.
```

**Mistral7B + Llama3.1-8B, Stitched at 60%**

```
We are still working on the Homecoming planning
and need your help. We need to find a way to get
the Homecoming banners and the Homecoming flags.
```

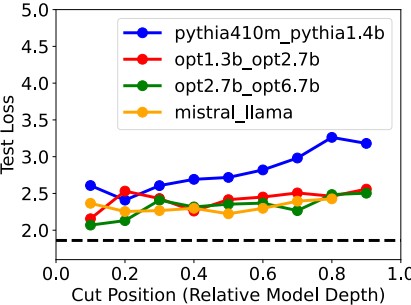

Figure 1: (Left) Text generation examples of several stitched models, and (right) test loss as a function of stitched position, where two models are cut at the same position relative to their model depth. The black dashed line on the right figure indicates the average test loss of original models.

nearest-neighbor set. For computational tractability, we set $\epsilon = 0$ for our LLM-LLM stitching, while allowing $\epsilon \neq 0$ for the other cases such as LLM-MLP stitching.

Formally, our process of stitchng two LLMs is as follows: we consider two LLMs $\mathbf{A} = \mathbf{U}^A \left( \prod_{i=0}^{n-1} \mathbf{H}_i \right) \mathbf{E}^A$ and $\mathbf{B} = \mathbf{U}^B \left( \prod_{i=0}^{m-1} \mathbf{K}_i \right) \mathbf{E}^B$, where $\mathbf{H}_i, \mathbf{K}_i$ are decoder layers, $\mathbf{E}$ is the embedding layer, and $\mathbf{U}$ is the unembedding layer. The stitched model is given by

$$\mathbf{B} \circ \mathbf{A} = \mathbf{U}^B \left( \prod_{i=(m-l+1)}^{m-1} \mathbf{K}_i \right) S(\Lambda) \left( \prod_{i=0}^{k-1} \mathbf{H}_i \right) \mathbf{E}^A, \tag{1}$$

where we stitched the first $k$ layers of $\mathbf{A}$ and the last $l$ layers of $\mathbf{B}$. We then train a linear stitching layer $S(\Lambda)$ to minimize the next-token prediction cross-entropy loss:

$$\mathcal{L}(\Lambda) = \sum \log \left[ \mathbb{P}(v_i | v_{i-1} \cdots v_0, \Lambda) \right]. \tag{2}$$

For stitching models with different tokenizers, $v_i$ is the first token of the string $v_i v_{i+1} v_{i+2} \cdots$, tokenized by $\mathbf{B}$'s tokenizer.

## 5 UNDERSTANDING ALIGNMENT OF LLM REPRESENTATIONS VIA STITCHING

Could we stitch two different LLMs to obtain another LLM whose performance is comparable to original models? Specifically, we consider (a) stitching LLMs of different sizes within the same model family, and (b) stitching two LLMs from different model family. Note that models from different model family typically have different tokenizers.

We use OPT–{1.3B, 2.7B, 6.7B}, Pythia–{410M, 1.4B, 2.8B}, Mistral-7B-Instruct, and LLaMA-3.1-8B-Instruct for the stitching experiment. We used open-source models available in Huggingface, and used Huggingface datasets *monology/pile-uncopyrighted* and *monology/pile-test-val* for training and evaluating test loss. Each sample was truncated at the 2048th token, and 2000 randomly chosen samples from the test set were used to evaluate the test loss.

Figure 1 shows the result of stitching LLMs. We found that stitching is possible across a range of stitching positions, with text generation capabilities slightly worse than the original model, as measured by the test loss.

To evaluate the equivalence of representations at different stages of model inference, we also experimented stitching various layers of one model onto a fixed layer of another model. Results are shown in Figure 2 and 3. We observe that the embedding layer is extremely hard to stitch to any other parts

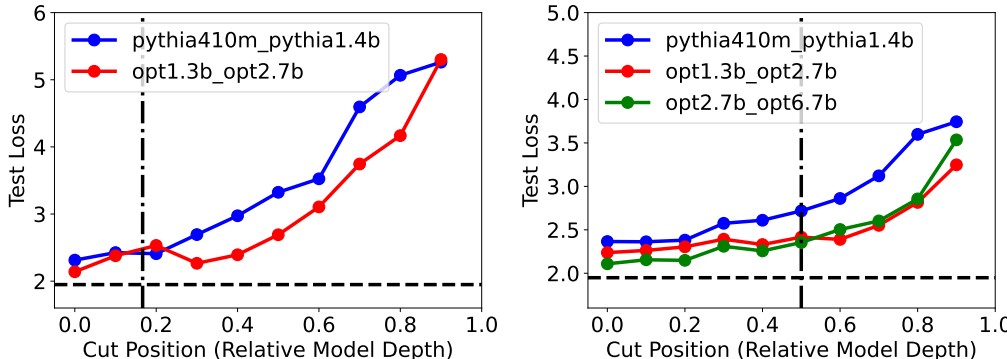

Figure 2: Test loss as a function of stitched positions, where layers of the second model are stitched onto (left) 1/6-depth layer, and (right) half-depth layer of the first model. The vertical dashed line indicates the relative position within the first model onto which the second model's layers are stitched, and the horizontal dashed line indicates the average test loss of original models.

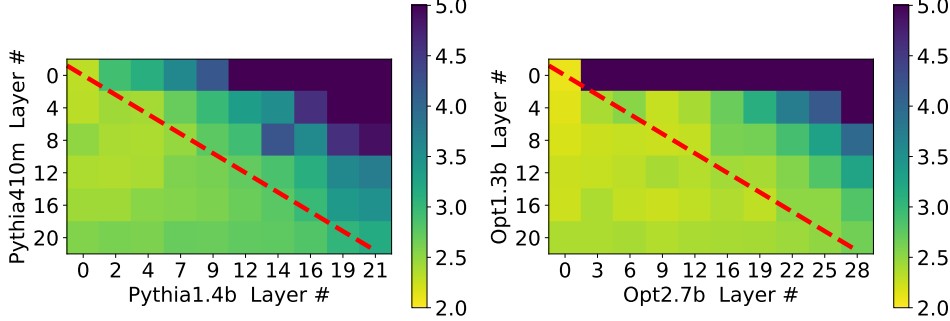

Figure 3: Test loss as a function of stitched positions, where the first part of the y-axis model is stitched onto the second part of the x-axis model. The red diagonal line corresponds to stitching layers at the same relative depth within each model.

of the model, which implies that there are catastrophic changes in representations during the early few layers to convert token-level representations to useful concepts. Mid-layer representations could be reasonably stitched to other parts of the model, while the stitching performance becomes worse as it's stitched to later layers of the model. This implies that at later layers, the model starts to focus on next-token predictions rather than forming high-level concepts, making them pretty different from mid-layer representations. It is also worth noting that the mid-layer representation could be stitched to early layer of another model (*e.g.* connecting layer 0-15 of Pythia410m with layer 2-23 of Pythia1.4b), which implies that mid-layer representations still include some token-level information, and it is possible to reset the representation back to token-only representations.

The overall stitching result indicates that LLMs of different scales and architectures may follow similar stages of knowledge processing, *i.e.*, forming semantic concepts around early-mid layer and starting to generate next token predictions at late layers, aligning with proposals made in the literature (Lad et al., 2024).

## 6 EXPLORING UNIVERSALITY: A GLIMPSE INTO THE GEOMETRIC STRUCTURE OF LLM REPRESENTATIONS

In this section, we take a further glimpse into the geometric structure of LLM Representations by stitching LLMs to MLPs and hand-crafted decoders, using family tree representations as an example. We will find intriguing hints of universality whereby equivalent representations are learned in these

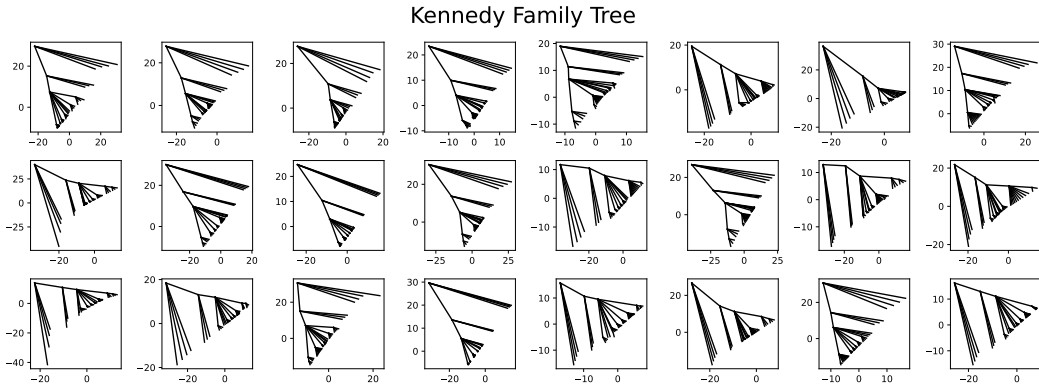

Figure 4: Kennedy family tree graph embeddings learned after different random initializations are found to be equivalent. We added straight lines connecting direct parent-child links for visualization.

very different contexts. We took the residual stream activations at the last token of *PersonA* in the following prompt:

**System prompt:** Recall the [*context*] family tree. You will be asked to answer a question about [*ancestor-descendant / family*] relationships in the [*context*] family tree.

**User prompt:** Is [*PersonA*] [*Relation*] of [*PersonB*]?

Here, *context* ∈{American Kennedy, Indian Nehru-Gandhi, European Rothschild}; In the main text, we focus on studying representations of Kennedy family tree. We also analyze how LLMs represent the family-tree in-context, where we provide a synthetic family tree as a context, before asking a question about the relation between two people. An example of in-context prompt is as follows:

**In-context prompt:** Consider the following synthetic family tree. You will be asked to answer a question about an ancestor-descendant relationship in the given family tree. Paul is a parent of John. John is a parent of Eric. [. . . ] Nicholas is a parent of David. Is [*PersonA*] a direct ancestor of [*PersonB*]?

We start by analyzing MLP representations of the Kennedy family tree. The learnable parameters in this case are the embedding vectors $\mathbf{E}_i$ as well as the weights and biases in the MLP, which defines the predictor function $\mathbf{p}(\mathbf{E}_i, \mathbf{E}_j)$. We provide training details in Appendix A. When the model is trained to learn *descendant-of* relationship, we found that the models trained with different random initializations converge to 2D *cone* embeddings, as shown in Figure 4. From these images, one could infer the machine-learned universal decoder algorithm: $j$ is a direct descendant of $i$ iff $\mathbf{E}_j$ lies within a fixed cone emanating from $\mathbf{E}_i$. Moreover, we could construct a reference decoder where the cone boundaries are exactly vertical and horizontal:

$$\mathbf{p}(\mathbf{E}_i, \mathbf{E}_j) = \sigma(\mathbf{E}_{i,1} - \mathbf{E}_{j,1})\sigma(\mathbf{E}_{i,0} - \mathbf{E}_{j,0}), \tag{3}$$

where $\sigma$ is a sigmoid function with trainable width. Figure 4 shows MLP embeddings which are learned from different initializations, but achieve near-perfect accuracy when fed into the reference decoder after a suitable AAT. We observe similar hints of universality for trees with different topology as well. We provide additional figures in the Appendix (Figure 10).

When we train the neural network to learn 18 different family relations: {*father, mother, husband, wife, son, daughter, brother, sister, grandfather, grandmother, aunt, uncle, nephew, niece, brother-in-law, sister-in-law, ancestor, descendant*}, we observe that these embeddings develop common features such as generation index and gender, regardless of their initialization. The table of equivalence scores between embeddings learned after different initializations, often shows strikingly high equivalence, as shown in the Appendix (Figure 11).

We now study LLMs' representations of the Kennedy family tree. We first compare the representations of different LLMs, and locate the position where they develop semantic understanding of the tree, beyond just processing the tokens. Figure 5 shows the CKA (Centered Kernel Alignment) between Kennedy family tree representations of different models at different layers. We observe

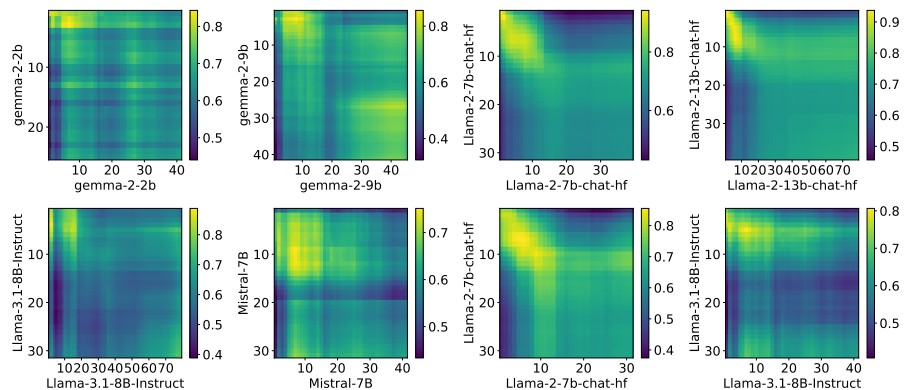

Figure 5: CKA (Centered Kernel Alignment) between Kennedy family tree representations of different models at different layers, where x and y axis represent layer index.

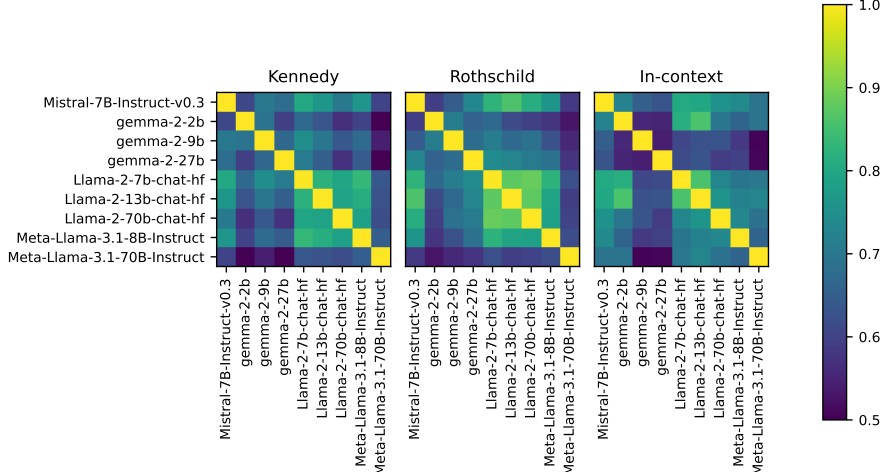

Figure 6: CKA (Centered Kernel Alignment) between various family tree representations of different models at 1/3 depth of the model. In-context family tree has the same topology as the Kennedy family tree.

that representations tend to be more similar at early-mid layers where models develop semantic concepts, and that the representations tend to be more similar between models of similar scale. Figure 6 shows the table of representation similarity between models at 1/3 depth of the model, where we could identify some blocks of similar representations. For instance, most Llama models have similar representations. Measuring representation alignment between layers within the same model, shown in Figure 7, supports the two-stage knowledge processing hypothesis of LLMs: over the progress of layers, most models have largely two blocks of similar representations, each corresponding to (a) forming semantic concepts, and (b) predicting next tokens.

We now compare the LLM representations, and geometrically interpretable MLP representations. For this, we first prune the Kennedy family tree to extract a part of the tree that the model knows: We do this iteratively starting from the root node, visiting nodes in breadth-first search order. For each visited node $v$, we ask questions to LLM about the relations between $v$ and all nodes that are in the current list. If the LLM gets all positive (*i.e.* Correct answer is Yes) questions correct, and more than 80% of the entire questions correct, we add the node $v$ to the list. Otherwise, we continue until we visit all nodes in the original tree. Using the pruned tree, we measure AAT-ES between the first 10 principal components of LLM representations and MLP. The result is shown in Figure 8, indicating that Kennedy family tree representations from Llama-3.1-70B LLM weights and LLM in-context are seen to be more universal than random embeddings.

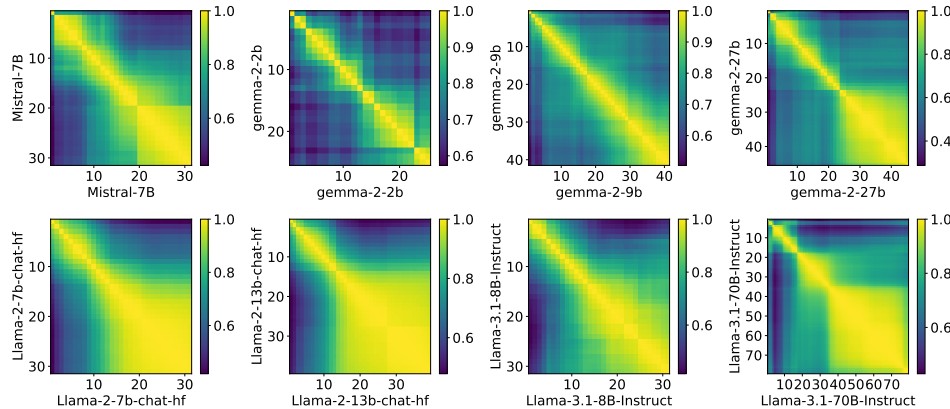

Figure 7: CKA (Centered Kernel Alignment) between Kennedy family tree representations at different layers within the same model, where x and y axis represent layer index.

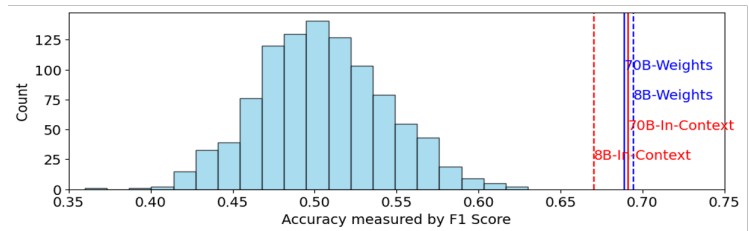

Figure 8: Kennedy family tree representations from Llama-3.1-70B LLM weights and LLM in-context are seen to be more universal than random embeddings, shown as histograms.

We also observe that LLMs oftentimes encode gender as their top principal component, as well as some generation index information in the following principal components, which further supports hints of universality. We have not yet been able to understand how neural networks process complicated family relations, such as brother-in-law or uncle. LLMs typically show bad performance in answering these questions about complex family relations as well, when they are not allowed to do chain-of-thought (CoT) reasoning. We believe understanding how LLMs process representations during CoT is an interesting research question worth pursuing.

## 7 STARVATION HYPOTHESIS: UNIVERSALITY OF REPRESENTATIONS

Figure 9 shows the *Goldilocks* zone in the genealogical tree learning. When the decoder's size is too small, it cannot even fit the training data well, let alone generalize to unseen test data. When the decoder's size is too large, the model overfits and fails to generalize: while the training accuracy remains nearly 100%, the test accuracy drops. The plots suggest that the generalization occurs from starvation, *i.e.*, the necessity to discover generalizable and universal representations within the limited number of trainable weights.

More generally, if the decoder is too dumb, it never outperforms chance. If the decoder is too smart, it can simply memorize (overfit) the data. Either way, the encoder loses its incentive to learn a clever compact representation. In the Goldilocks zone, however, the decoder is incentivized to learn the most clever representation, defined as the one that's easiest to decode. Starvation is presumably caused not by small overall resources, but by fierce competition for resources against other LLM tasks. In today's world with ever-growing amount of data, our LLMs will never be capable enough to memorize all the data, so they are incentivized to learn the most clever (hence universal) representation, compressing the data in a way that allows generalization.

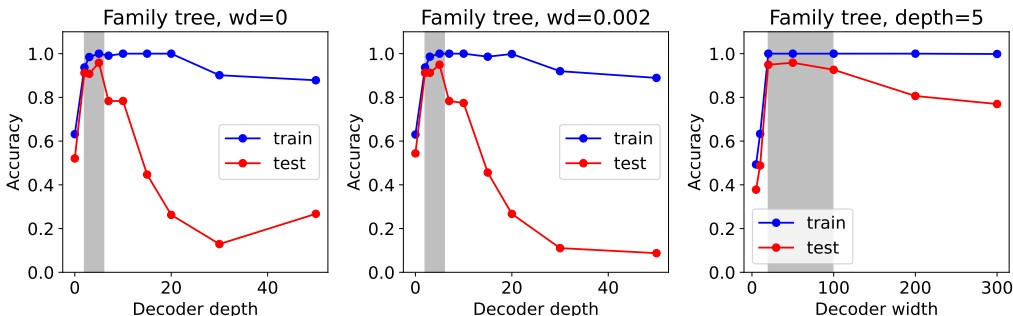

Figure 9: Plot of train/test accuracy as a function of MLP decoder's depth, width, and weight decay, indicating *Goldilocks zone*. Default decoder width was set to 50 and 75% of the data was used for training.

## 8 CONCLUSION

In this paper, we find hints of universality, where equivalent representations are learned across a range of model sizes (from $10^2$ to $10^9$ parameters) and contexts (MLP toy models, LLM in-context learning and LLM training). We find experimental support for such universality by showing that LLMs and simpler neural networks can be stitched, i.e., by stitching the first part of one model to the last part of another, mediated only by an affine or almost affine transformation. We hypothesize that this dynamic toward simplicity and generalization is driven by "intelligence from starvation": where overfitting is reduced by pressure to minimize the use of resources that are either scarce or competed for against other tasks.

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

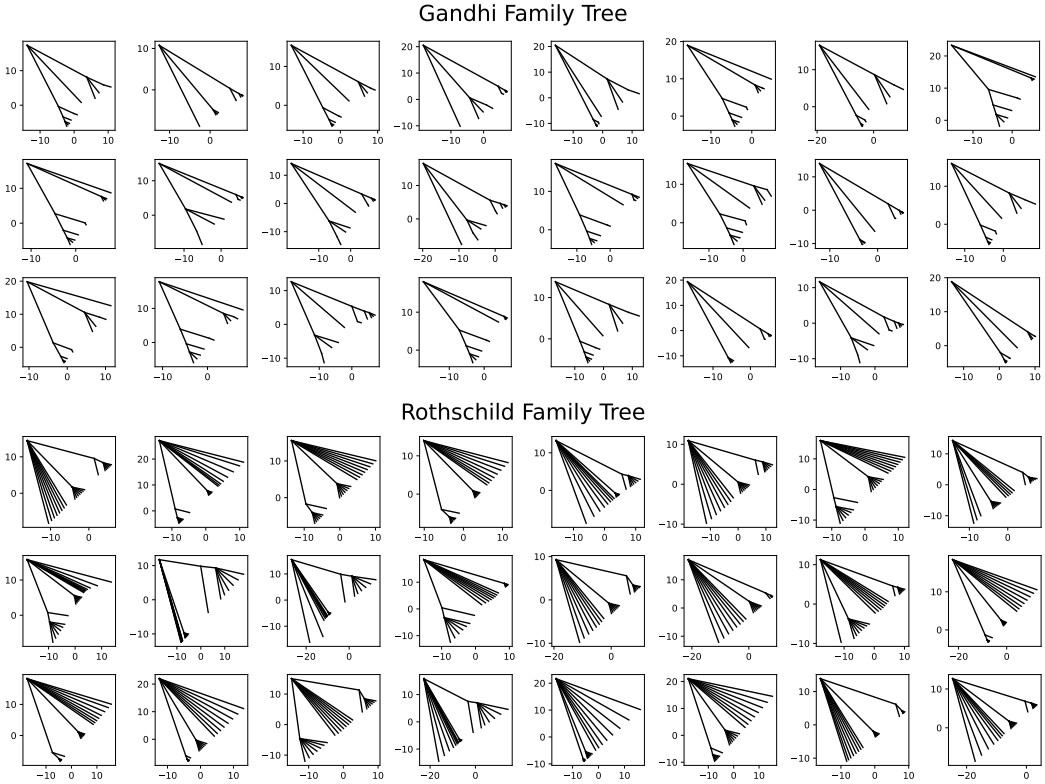

Figure 10: Family tree graph embeddings of MLP which is trained to learn *descendant-of* relationship. We added straight lines connecting direct parent-child links for visualization.

## A    TRAINING DETAILS OF MLP FOR LEARNING GENEALOGICAL TREE

We trained full/descendant-only family trees with 200 different initializations. We trained an embedding and the MLP decoder for 3000 steps. We then analyzed the representations of those that achieved 100% accuracy within 3000 steps. Figure 10 shows additional family tree graph embeddings of MLP for different family trees. Figure 11, 12, and 13, illustrate the MLP representations trained to learn the entire set of family relations, indicating high equivalence scores between embeddings learned after different initializations. Most representations develop common features such as generation index and gender.

Table 1: Table illustrating the number of different initializations that passed each criterion.

| Family Tree Data | Kennedy desc | Kennedy full | Gandhi desc | Gandhi full | Rothschild desc | Rothschild full |
|---|---|---|---|---|---|---|
| 100% Train Acc | 123 | 52 | 199 | 194 | 190 | 85 |
| $\geq$ 99% Acc w/ Ref Decoder | 57 | – | 171 | – | 124 | – |

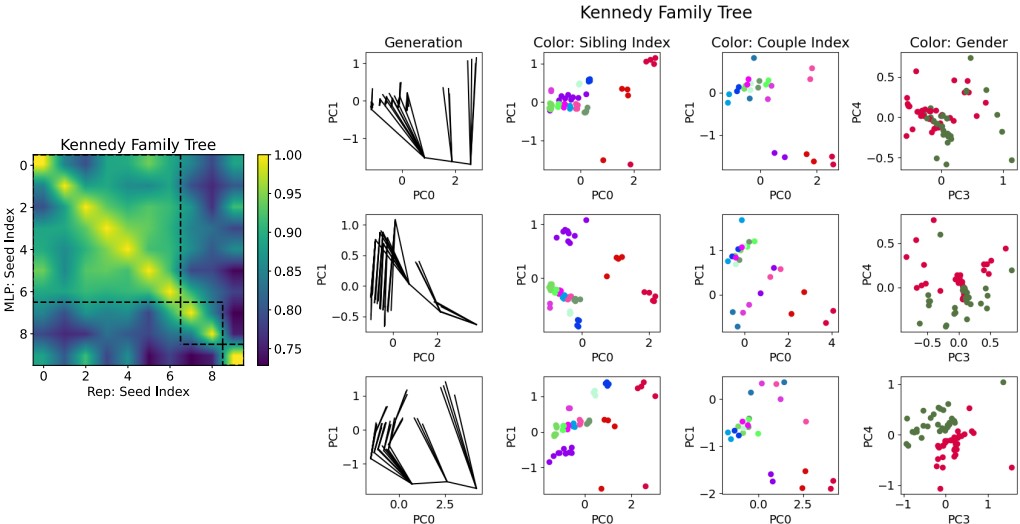

Figure 11: (Left) Heatmap showing AAT-ES, where $(i, j)$ cell shows the Equivalence score by feeding $i$-th representation into $j$-th MLP decoder for learning the full Kennedy family tree. (right) Each row shows several PCA plots colored by different information for each random seed. These plots indicate that Genealogical tree learning in MLP develops common features such as generation index, siblings index, couple index, and gender.

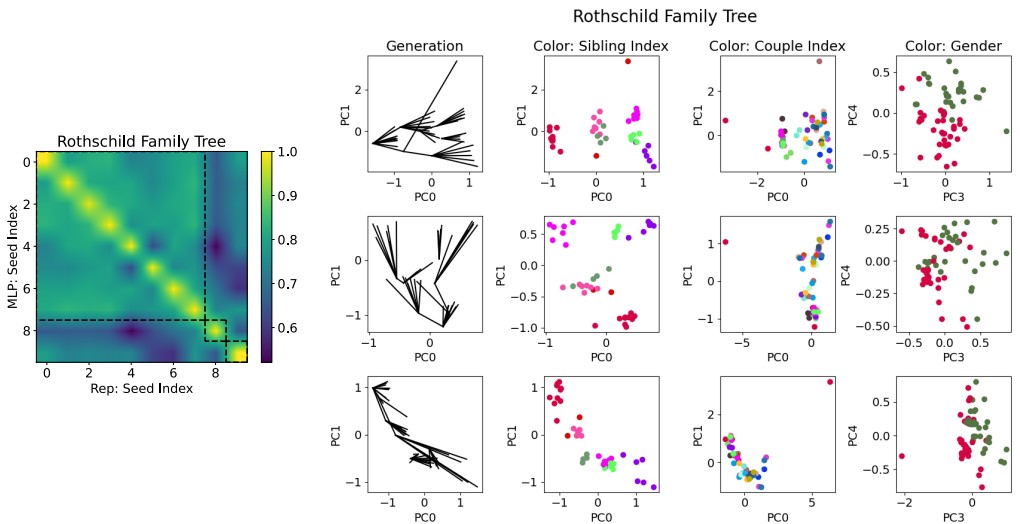

Figure 12: (Left) Heatmap showing AAT-ES, where $(i, j)$ cell shows the Equivalence score by feeding $i$-th representation into $j$-th MLP decoder for learning the full Rothschild family tree. (right) Each row shows several PCA plots colored by different information for each random seed. These plots indicate that Genealogical tree learning in MLP develops common features such as generation index, siblings index, couple index, and gender.

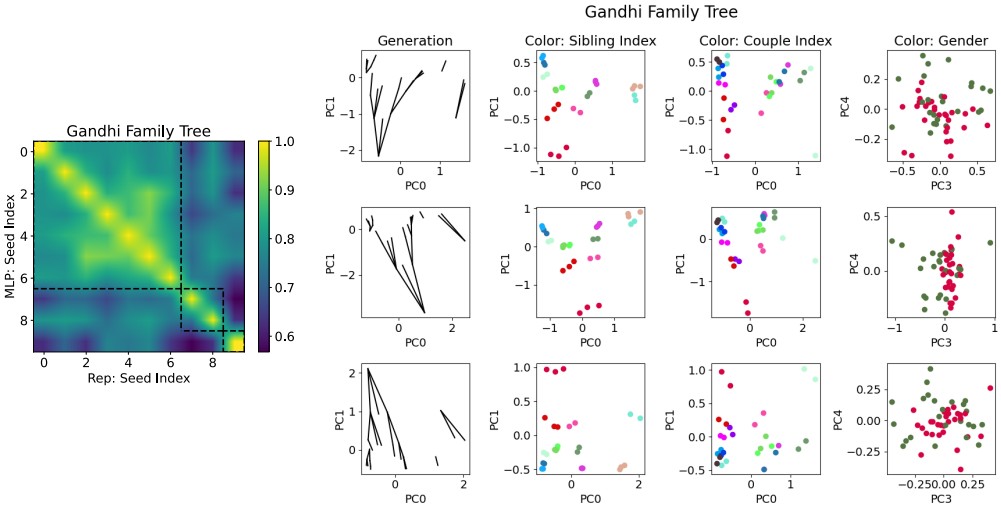

Figure 13: (Left) Heatmap showing AAT-ES, where $(i, j)$ cell shows the Equivalence score by feeding $i$-th representation into $j$-th MLP decoder for learning the full Nehru-Gandhi family tree. (right) Each row shows several PCA plots colored by different information for each random seed. These plots indicate that Genealogical tree learning in MLP develops common features such as generation index, siblings index, couple index, and gender.

