# OpenReview forum: "Generalization from Starvation: Hints of Universality in LLM Knowledge Graph Learning"
_ICLR.cc/2025/Conference — ICLR 2025 Conference Withdrawn Submission_

### Official Review · Reviewer_h8CR · 2024-10-29

**Soundness:** 1
**Presentation:** 2
**Contribution:** 2
**Rating:** 3
**Confidence:** 3

**Summary:**

This study examines similar representations uncovered across a variety of models trained on a graph dataset. It also studies how representations may transfer by "stitching" together different models at different layers, inserting a trainable (approximately) linear layer between them.

**Strengths:**

The topic of study is quite interesting, and sounds like a limited version of the "Platonic Representation Hypothesis" (https://arxiv.org/abs/2405.07987). Investigating how different learned representations are similar or different across various settings and models is a fascinating and important topic.

**Weaknesses:**

The paper overall seems to be unfinished, and makes a variety of vague, unusual, or incomplete claims:
- The abstract refers to the representations under study as "attractor" representations, but there do not seem to be any dynamics under investigation
- Line 152: what are "superfluous nonlinearities" and why do they need to be absorbed? How do you choose $\varepsilon$?
- Line 187: These expressions seem to suggest your models are feedforward with no nonlinearities or attention?
- Section 5: is there a way we can judge the scale of the loss? What constitutes a bad loss? Your stitched models seem to perform worse, but how can we quantify how much worse this is?
- Fig 4: How are these embeddings visualized? Do you use PCA?
- Lines 370 - 377: why do you prune the data graph to keep only successful judgements, rather than using the whole graph? Wouldn't this inflate the accuracy of your models you then measure in Fig. 8? I'm not entirely sure what's being measured here, either. How is this comparing LLMs to MLPs?
- Fig 8: Certainly, your trained models do somewhat better than the random embeddings, but is 65-70% accuracy on a yes/no task considered "good" in this setting? Part of my confusion may be that I'm unsure what you're trying to measure in this plot.
- Section 7: How is "starvation" different from traditional descriptions of overfitting/underfitting, e.g. through bias-variance decomposition? This discussion seems anthropomorphize LLMs to an unusual extent, describing their components as being "too dumb" or "too smart," engaged in a starvation-fueled fierce competition for resources. While such description are fine for conveying intuition, they are also imprecise, and it's unclear what the authors are attempting to communicate.

**Questions:**

See weaknesses above.

---

### Official Review · Reviewer_QVmD · 2024-11-01

**Soundness:** 1
**Presentation:** 2
**Contribution:** 2
**Rating:** 3
**Confidence:** 4

**Summary:**

The paper provides empirical evidence that representations of hierarchical graphs (genealogical trees) in LLMs and MLPs are similar through both model stitching and CKA (centered kernel alignment) analyses.

**Strengths:**

The effort to understand how knowledge graphs are represented across networks, and the experiments quantifying loss after model stitching are the key strengths of this paper.

**Weaknesses:**

Unfortunately, the weaknesses of the paper significantly outweigh its strengths. The paper repeatedly promises to provide experiments and details which are never provided. No clear evidence is provided for any of the following claims:
- Representations optimize generalization performance.
- Representations exploit properties of the knowledge graph relations to achieve optimality.
- Universal representations allow for generalization.
- Representations are provably optimal.

The omission of details regarding experiments and analysis hampers understanding of results and conclusions, most critically when the “starvation hypothesis” is presented in Section 7. Section 7 is particularly unclear, seemingly interchanging the roles of encoder and decoder and making claims that are not supported by the provided results.

Results for representations generated by in-context prompts are at times provided but never discussed, despite the “starvation hypothesis” making important claims about how generalization performance should relate to universality of representations.

The general formulation presented in Section 3 is never utilized.

There are errors in figure captions, references, axes labels, and data presentation that hamper the ability to assess the paper.

**Questions:**

I believe this paper requires significant modifications to be of publishable quality.

---

### Official Review · Reviewer_27g1 · 2024-11-03

**Soundness:** 1
**Presentation:** 2
**Contribution:** 2
**Rating:** 3
**Confidence:** 4

**Summary:**

This paper studies the notion of universality in language models. It addresses this question using different experiments. First the paper shows that language models of different sizes can be merged together by merely training a linear transformation (setting eps=0.0) to translate the representations. The paper then investigates a knowledge graph retrieval task where relational questions are asked on a family graph. The authors claim from these experiments: 1) Models of similar size have similar representations 2) Models have two blocks of representational similarity corresponding to forming semantics and next token prediction. 3) LLM representations are more aligned to a trained MLP compared to random representations. The paper closes by proposing that starvation of resources forces compact and universal representations.

**Strengths:**

1: This paper introduces almost affine transformations as a measure of representational similarity.

2: This paper investigates universality, an important conceptual question to consider as we train big models on a large corpus of data.

3: This paper demonstrates that LLM representation from weights and in-context show a degree of universality closer to a trained MLP than random representations.

4: In general, this paper proposes the \textit{right} experiments which are needed to understand universality.

**Weaknesses:**

Albeit the importance of the topic of study, the paper has major flaws in all 1) the rigorousness of the experiments 2) description of experimental details 3) how conclusions are drawn from resulting plots 4) introducing seemingly novel but previously known concepts (In decreasing order of importance). W7, W6, W8, W2, W1 are the main reasons justifying the score (in decreasing order of importance).

W1: It is unclear how successful model stitching was. Figure 1 shows that the test loss goes from 2 to 2.5 or 3, but looking at Hoffman et al. (https://arxiv.org/abs/2203.15556)’s online training loss curve, this amount of change in loss is more than 10^21 FLOPs (The majority of the pre-training compute) for ~1B parameter models. I think how well model stitching worked should be demonstrated beyond showing 3 generation examples. This applies to every part of Section 5. Without a demonstration to what extent ICL abilities (e.g. Winograd, ARC1d) and general knowledge (MMLU) abilities are preserved, a loss change of 2 to 3 seems like a pretty significant one. This claim is a big claim towards universality, and thus requires a substantial amount of rigor.

W2: Causal and mechanistic findings are drawn just from CKA plots. First, I believe there isn’t enough evidence to support line 362, “ the representations tend to be more similar between models of similar scale”. The pattern of representational similarity between layers is similar for models of similar scale, but Figure 5 doesn’t support the statement in line 362. Second, Line 367 “most models have largely two blocks of similar representations, each corresponding to (a) forming semantic concepts, and (b) predicting next tokens.” also seems to be not well justified. This conclusion indeed seems *plausible* from Figure 7, but there isn’t any experiment directly justifying this hypothesis (e.g. activation patching).

W3: It is unclear why the known part of the KG is extracted using the given pipeline,  only for Figure 8, and not for Figure 6,7.

W4: Is comparing AAT-ES to a random representation enough to claim evidence towards universality? Perhaps at least a control where a different graph structure is used might be more fair. As an analogy, is it surprising that similar edge detectors are found for networks trained on ImageNet and COCO as opposed to random features?

W5: The paper is written with many oral language terms and newly introduced words, which do not reflect the terms already present in the literature. Line 425 “dumb”, can be either replaced with “not expressive”, or “under-optimized” (if it is an optimization problem). I think using easy language is fine, but it should deliver an accurate meaning. The terms “goldilocks zone” or “starvation hypothesis” are also not quite needed to be introduced as novel concepts, as will be discussed in the next point. The paper seems to be coining new terms for existing phenomena.

W6: The goldilocks zone or the starvation hypothesis is not novel as we already know that traditional overfitting curves already demonstrate this phenomena. These effects are not only discussed in traditional statistical learning theory [1,2] but also in the context of neural networks [3,4]. These are explicit examples, while in general the notion of “forcing to learn compact representations”, has always existed throughout machine learning research as regularization [5], data augmentation [6], weight decay [7], dropout [8], sharpness aware optimization [9], etc. Nobody has done exactly the experiments discussed here, however, it isn’t clear from the paper what makes this finding novel compared to the vast literature above. The authors should have at least cited and discussed such works.

W7: Most of the findings lack a quantification of uncertainties or sensitivity to hyper parameters. e.g. In Appendix A: “We trained an embedding and the MLP decoder for 3000 steps. We then analyzed the representations of those that achieved 100% accuracy within 3000 steps.” This doesn’t describe the model input/output, tokenization scheme, initialization, optimizer, batching, regularization, etc. The sensitivity/convergence within 3000 steps should also be analyzed. I find these analyses extremely important as the effect of model size is often confounded with simply faster training, as wider models train faster given the same # of gradient steps.

W8: It is unclear why AAT was introduced but not used in the first KG experiments (Figure 5,6,7). The AAT introduction section clearly mentions that this metric is better suited for measuring representational similarity, but this has not been demonstrated. It isn’t clear when and why CKA/AAT-ES is used.

W9: Some terms are ill defined and many part of the experiments lack detail (see questions).

[1] https://arxiv.org/abs/2004.04328
[2] https://arxiv.org/abs/2303.14151
[3] https://arxiv.org/abs/1912.02292
[4] https://arxiv.org/abs/2310.18988
[5] Cortes, Corinna, Mehryar Mohri, and Afshin Rostamizadeh. "L2 regularization for learning kernels." arXiv preprint arXiv:1205.2653 (2012).
[6] Shorten, Connor, and Taghi M. Khoshgoftaar. "A survey on image data augmentation for deep learning." Journal of big data 6.1 (2019): 1-48.
[7] Krogh, Anders, and John Hertz. "A simple weight decay can improve generalization." Advances in neural information processing systems 4 (1991).
[8] Srivastava, Nitish, et al. "Dropout: a simple way to prevent neural networks from overfitting." The journal of machine learning research 15.1 (2014): 1929-1958.
[9] Foret, Pierre, et al. "Sharpness-aware minimization for efficiently improving generalization." arXiv preprint arXiv:2010.01412 (2020).

**Questions:**

Questions:

1. Why do some experiments use AAT-ES and some CKA?

2. How are: “representations from Llama-3.1-70B LLM weights and LLM in-context” defined? I guess they are both activations? What exactly is the context?

3. How exactly is AAT-ES trained? Could the authors list out hyperparameters and the data size used to train the method?

Suggestions:

1. Figure 2’s caption could be improved, I had to read the caption multiple times to understand what the experiment was.

---

### Official Review · Reviewer_FRv2 · 2024-11-04

**Soundness:** 1
**Presentation:** 1
**Contribution:** 2
**Rating:** 3
**Confidence:** 3

**Summary:**

The paper studies the hypothesis that llms learn representations that are transferable and hence universal across model sizes and families. The authors use stitching with affine or quadratic bridge functions between two models to conduct their study, as well as some visualization of representations learned in feed-forward neural nets trained on predicting family relations. Finally, the authors conjecture that such universal representation emerge because of "starvation" and competition for resources.

**Strengths:**

- stitching different model representations (if further developed and investigated) might be an interesting tool to understand how aligned various models are.
- the idea that representations learned by (some classes of models) are "universal" is thought provoking, although I believe some tighter scoping and more precise definitions would make the claim easier to follow.

**Weaknesses:**

I had hard time following this paper; the structure of the presentation could be improved, moving elements closer to where they are used and being more focused about what I believe to be the main contribution: i.e. introducing stitching and a related "equivalence score".
- The paper suggests that it investigates representations related to knowledge graph learning (KGL), but it seems that the only real experiment about learning is conducted on small MLPs. Unless I have missed some point, there is no experiment in which an LLM is trained to explicitly solve a KG task.
- Definitions and discussion on page 3 about "optimal KG representations" and other properties seem to have not much bearing in the rest of the work. Where are they used? Does the proposed similarity metric behaves well in this regard of symmetry and equivalence?
- Definition 1. and surrounding about equivalence score is very unclear. What are the input and output? How are the domain and codomain defined? When are non-linearities "superfluous"?
  - A formal definition of equivalence score is missing. A formal definition would make clear what is required to compute the proposed metric (e.g. a dateset, a loss function, an optimization routine, ...) which at the moment remains implicit in the presentation;
   - A study of properties of the proposed score is also missing. E.g. is the score invariant to some transformations?
   - In fact, I think the paper would benefit also from a clarification and possible definition of universality. When is a representation indeed universal?
- After definition 1 the authors say that AAT-ES is to be preferred to CKA, but then they actually report CKA in many experiments, eg. Fig 5 and 6.
     - also, in which sense is CKA a "nearest-neighbor" measure? CKA is closely related to CCA and linear regression, which I don't see how they are nearest-neighbor measures. Can you point me to specific passages of the CKA paper (or any follow up work) that support this statement?
- It is difficult to put any study about test loss via stitching (Fig 1, 2, and 3) in perspective, without any reference test loss (e.g. how bad is a test loss of 2.5 vs 3?). Also, the work completely misses some "sanity check" experiment, e.g. what happens when stitching together random vectors (e.g. representation obtained through a randomly initialized token embedding layer) with an LLM? I would expect that since the stitching layer may work as a sort of token embedding layer, you may be able to obtain reasonable final output and "test loss".
- I could not understand Figure 8. What is the count about?
- I find the experiment leading to Figure 9 unsurprising, it simply depicts a standard underfitting-overfitting scenario for MLPs; I do not see how this study would reasonably support a "starvation hypothesis" as it is presented in the paper; especially how does the MLPs example transfer to LLMs that contain orders of magnitude more parameters and are trained on orders of magnitude more data. Where would an LLM such as the various versions of Llama 3 stand in the plot?
   - Also how does Figure 9 relates to studies about scaling laws and double descent scenarios?

Finally, I would advise the authors to thoroughly check the work for style and mathematical notation.

**Questions:**

See weaknesses.

---

### Note · Authors · 2024-11-26

**Comment:**

We'd like to thank the reviewers for their comments and suggestions. Clearly, we have some work to do on improving the presentation of the paper and on clarifying its contributions, and so we are withdrawing the paper from consideration at ICLR 2025.

**Withdrawal Confirmation:**

I have read and agree with the venue's withdrawal policy on behalf of myself and my co-authors.